# Current Status of and Future Perspectives in Bacterial Degradation of Benzo[a]pyrene

**DOI:** 10.3390/ijerph18010262

**Published:** 2020-12-31

**Authors:** Alexis Nzila, Musa M. Musa

**Affiliations:** 1Department of Life Sciences, King Fahd University of Petroleum and Minerals, Dhahran 31261, Saudi Arabia; 2Department of Chemistry, King Fahd University of Petroleum and Minerals, Dhahran 31261, Saudi Arabia; musam@kfupm.edu.sa

**Keywords:** benzo[a]pyrene, biodegradation, co-metabolism, bioaugmentation, catabolic pathways, omics, functional metagenomics

## Abstract

Benzo[a]pyrene (BaP) is one the main pollutants belonging to the high-molecular-weight PAHs (HMW-PAHs) class and its degradation by microorganisms remains an important strategy for its removal from the environment. Extensive studies have been carried out on the isolation and characterisation of microorganisms that can actively degrade low-molecular-weight PAHs (LMW-PAHs), and to a certain extent, the HMW-PAH pyrene. However, so far, limited work has been carried out on BaP biodegradation. BaP consists of five fused aromatic rings, which confers this compound a high chemical stability, rendering it less amenable to biodegradation. The current review summarizes the emerging reports on BaP biodegradation. More specifically, work carried out on BaP bacterial degradation and current knowledge gaps that limit our understanding of BaP degradation are highlighted. Moreover, new avenues of research on BaP degradation are proposed, specifically in the context of the development of “omics” approaches.

## 1. Introduction

Polycyclic aromatic hydrocarbons (PAHs) represent a class of pollutants that are ubiquitous in the environment. They are present in crude oil, thus they contaminate the environment as the result of oil exploitation, transport and refinement. They also have an anthropogenic source, as the result of the incomplete burning of coal, wood, oil, tobacco, and meat cooking at high temperature [1,2]. PAHs are classified into low-molecular-weight PAHs (LMW-PAHs), consisting of two or three fused rings such as naphthalene, phenanthrene, anthracene, and fluorene; and high-molecular-weight PAHs (HMW-PAHs), consisting of four or more rings such as fluoranthene, pyrene, benzo[a]anthracene, and benzo[a]pyrene (BaP) [1,2,3].

Benzo[a]pyrene, which consists of five fused aromatic rings, is one of the most interesting PAHs. The presence of these rings confers this compound high stability against biodegradation. Moreover, its low bioavailability makes it a less preferable a source of energy by microorganisms, due to the large amount of energy needed to break carbon–carbon bonds [1,2,4]. Therefore, BaP tends to accumulate in the environment, with its attendant toxicity [5].

In human body, BaP is a well-established carcinogen, and is primarily associated with the occurrence of lung cancer [6,7,8]. In animals, oral exposure to BaP has been shown to result, besides cancer, in developmental toxicity (including developmental neurotoxicity), reproductive toxicity, and immunotoxicity [9,10,11,12,13]. BaP has been classified by the US Environmental Protection Agency (USEPA) among the top ten priority pollutants of great concern [14]. Thus, its removal from the environment remains a priority.

Among the efficient strategies to remove PAHs in general, and BaP in particular, is the use of microbial degradation. This strategy rests on the exploitation of the ability of microorganisms to grow by using pollutants as their source of carbon, leading to partial or complete removal of these pollutants. Unlike physical and chemical approaches, biodegradation remains attractive since it is environmentally benign and cost effective [15,16]. In the current review, unless otherwise specified, the word “biodegradation” refers to “degradation by bacteria”.

A considerable amount of work has been dedicated on the biodegradation of PAHs [17,18,19,20]. A careful observation of these reviews shows that most of the reported work has been focused primarily on the biodegradation of PAHs containing two to four fused rings. Indeed, because of the recalcitrance of the BaP to biodegradation, little work has been carried out on this compound, in comparison with other PAHs. Nevertheless, reports are emerging on BaP biodegradation, and the current review summarises these reports. Recently, a review on BaP degradation has been reported, however, it has focused primarily on BaP degradation by fungi, and bacterial degradation was mentioned just in term of broader overview [21]. In the current review, work carried out on BaP bacterial degradation since the 1970s is presented, knowledge gaps that limit our understanding of BaP degradation have been highlighted, and biochemical degradation pathways are presented. This review also proposes new avenues of research on BaP degradation, specifically in the context of the development of “omics” approaches.

## 2. Studies on BaP Degradation

### 2.1. Degradation in Aerobic and Normal Conditions of Temperature and Salinity

The first report on bacterial degradation of BaP dates back to the mid 1970s. Before this time, it was known that BaP could be converted to active and/or toxic compounds by human cells, however, no available information on the possibility of BaP degradation by bacterial strains until the year 1975, during which, Gibson et al. reported the degradation of BaP by the mutant strain of *Beijerinckia* B-836 [22] (ST1, study 1 as in Table 1). In this study, this bacterial strain was first cultured in the presence of biphenyl and succinic acid, and then, when exposed to BaP, the dihydrodiol derivative *cis*-9,10-benzo[a]pyrene-dihydrodiol was detected; this is the first report on a bacterial metabolite of BaP. In the same year, the following strains, *Pseudomonas* NCIB 9816, *Pseudomonas* ATCC 17483, *Pseudomonas putida* PpG7, were reported to utilise BaP in the presence of succinate and salicylate [23] (ST2). In 1988, Heitkamp and Cerniglia investigated the mineralisation of various PAHs, including BaP, by a bacterial strain derived from pyrene enrichment of an oil contaminated sediment. Using ^14^C-radiolabeled PAHs and a mixture of growth substrates consisting of peptone, yeast extract and soluble starch, the biomineralisation (production of ^14^C-labeled CO_2_) of various PAHs was observed, except that of BaP. However, interestingly, this microorganism could generate BaP metabolites, as monitored in organic-extractable residues of the cultures [24] (ST3). This is an example of pollutant transformation versus biomineralisation, a concept discussed in Section 3.3. A further study showed that, the same microorganism, which was identified as *Mycobacterium* sp. PYR, could degrade BaP when combined with a contaminated sediment [25] (ST4a), an example of bioaugmentation (Section 3.3). This strain, *Mycobacterium* sp. PYR, was also used to investigate BaP metabolites, as discussed in Section 3.1.1 [26] (ST4b).

Compelling evidence of utilisation of BaP by bacteria was provided with the mineralisation of radiolabeled ^14^C-BaP to ^14^C-CO_2_ by a *Mycobacterium* sp. strain. This strain was initially isolated by enrichment experiments using soil contaminated samples from China in the presence of pyrene as a substrate [27] (ST5). Subsequently, and using the same strain, which was now named *Mycobacterium* RJGII-135, Schneider et al. confirmed the ability of this strain to mineralise radiolabeled ^14^C-BaP to ^14^C-CO_2_, [28] (ST6). Further investigations, using the same strain, led to the identification of BaP metabolites such as 7,8-benzo[a]pyrene-dihydrodiol (7,8-BaP-dihydrodiol) and those resulted from ring cleavage of BaP, including 4,5-chrysene-dicarboxylic acid and a dihydro-pyrene-carboxylic acid, providing the first evidence of BaP ring fission [28]. A mixture of bacteria, isolated from contaminated soil and consisted of *Pseudomonas putida*, *Flavobacterium* sp., and *Pseudomonas aeruginosa* was shown to degrade BaP, along with other PAHs, in the presence of glucose, yeast extract, and peptone as growth substrates [29] (ST7). In this study, it is interesting to note that the use of glucose inhibited the degradation of phenanthrene, a phenomenon known as metabolic inhibition (Section 3.2).

Ye et al. tested the ability of *Sphingomonas paucimobilis* strain EPA 505 to degrade HMW-PAHs, including BaP, in the presence of fluoranthene [30] (ST8). A resting cell suspension (1 mg wet cells/mL) of *S. paucimobilis* EPA 505 grown in the presence of fluoranthene could degrade 33% of BaP, after 16 h of incubation with 10 ppm of BaP. In this study, the biomineralisation of radiolabeled ^14^C-BaP to ^14^C-CO_2_ was also confirmed [30].

The degradation of BaP has also been reported in the presence of phenanthrene, using bacteria, including 3 *Burkholderia cepacia* strains isolated by enrichment experiments of soil contaminated samples [31] (ST9). Another study investigated radiolabeled and unlabeled forms of BaP in soil amended with mineral nutrients, in the presence of crude oil, by a soil microorganism consortium. Evidence of ^14^C-BaP conversion to ^14^C-CO_2_ was demonstrated. After 150 days, 95% of the tested ^14^C-BaP and 87% of crude oil were degraded, as measured by the quantification of the amounts of radiolabeled and unlabeled CO_2_ [32] (ST10). Mineralisation of ^14^C-BaP was also reported in the presence of phenanthrene on the following bacteria species: *Agrobacterium tumefaciens*, *Pseudomonas saccharophila*, *Pseudomonas* sp., *Burkholderia cepacia*, *Pseudomonas saccharophila*, *Bacillus cereus*, and *Sphingomonas paucimobilis*. These bacteria were isolated from several oil contaminated sites and enriched in phenanthrene-containing culture media [33] (ST11). One of the strains reported in the aforementioned study, *Pseudomonas saccharophilia* P15, was also shown to mineralise ^14^C-BaP to ^14^C-CO_2_ in the presence of salicylate [34] (ST12). The presence of salicylate leads to an induction of PAH dioxygenase, one of the enzymes associated with PAH degradation [34].

In a similar study, Kanaly and Bartha investigated the degradation of ^14^C-BaP in the presence of various substrates using soil microorganism consortium. These substrates consisted of individual hydrocarbons, defined hydrocarbon mixtures, crude oil, and crude oil fractions, including distillates heating oil, jet fuel and diesel. Overall, up to 60% of ^14^C-BaP degradation was observed with these substrates, however, the use of some co-substrates (such as 1-phenyldecane, pristine, naphthalene, phenanthrene and diesel vapor) was associated with inhibition of degradation [35] (ST13). This process, known as metabolic inhibition, is discussed in Section 3.2.

The first report on a bacterial degradation of BaP when used as a sole substrate was presented by Hunter et al. in 2005. In this study, the authors enriched a culture of a contaminated soil sample containing a mixture of pyrene, 1-aminopyrene, 1-hydroxypyrene and BaP, leading to the isolation of *Bacillus subtilis* Tgr3 strain [36] (ST14). The same study indicated that this strain could degrade BaP as a sole substrate in a minimum medium (MM). However, a careful observation of the experimental setup indicates that this MM could have contained another source of carbon (in addition to BaP), since in the control experiment, this strain grew when MM was used without BaP [36]. Thus, this may also be a case of co-metabolism, as discussed earlier.

In 2008, a *Sphingomonas yanoikuyae* JAR02 strain was shown to degrade BaP and mineralise it to CO_2_ in the presence of salicylate as a growth factor [4] (ST15). This strain, which was initially isolated by enrichment of a soil sample in the presence of phenanthrene, was unable to degrade BaP when succinate was used as a growth factor, a classical example of metabolic inhibition (Section 3.2).

The degradation of BaP has also been reported by *Stenotrophomonas maltophilia* (VUN 10,010), a bacterial consortium (VUN 10,009) and the fungus *Penicillium janthinellum* (VUO 10,201). These microbes were isolated in a gas plant-contaminated soil, and enriched in the presence of BaP and pyrene. When used separately, VUN 10,010, VUN 10,009 and VUO 10,201 could only degrade 10–15% of BaP (from 50 mg·L^−1^), and this rate increases to more than 40% when pyrene was used as a growth factor [37] (ST16). Interestingly, complete BaP mineralisation to CO_2_ was observed only in the context of co-culture of the fungus VUO 10,201 with either of bacterial strains. Thus, in this investigation, the fungus plays an important role in BaP minerilasation, in line with many previous works showing the ability of fungi to degrade BaP [58].

Another piece of evidence for BaP degradation by a single bacterial strain (and in absence of a growth substrate) was provided by Zang et al. in 2008. In this study, the authors investigated the ability of a *Zoogloea* sp. strain to degrade *cis*-4,5-BaP-dihydrodiol, *cis*-7,8-BaP-dihydrodiol and BaP, in the context of pre-oxidation of BaP using potassium permanganate (KMnO_4_) or hydrogen peroxide (H_2_O_2_). The study indicated that this strain could degrade around 60% of BaP within 15 days, from a mineral culture medium containing 50 mg·L^−1^), and this rate of degradation was further improved to more than 75% when a prior chemical oxidation of BaP was conducted. Similar results were obtained when *cis*-4,5-BaP-dihydrodiol and *cis*-7,8-BaP-dihydrodiol were used [38] (ST17). Another bacterial strain, *Bacillus subtilis* BMT4, was shown to degrade BaP when used as a sole substrate. This strain, which was isolated following the enrichment of contaminated soil samples in the presence of BaP, could degrade around 85% of BaP (50 mg·L^−1^), when used as a sole substrate, within 28 days of incubation [39] (ST18). This strain could degrade PAHs of lower molecular weight such as naphthalene and anthracene. The strain *Ochrobactrum* sp. BAP5 was also shown to degrade BaP as a sole substrate. It was isolated following an enrichment experiment of oil contaminated marine sediments, in the presence of BaP, and could degrade 20% of 50 mg·L^−1^ BaP within 30 days, [40] (ST19). Protein expression profile showed that the use of BaP was associated with the expression enzymes involved in the degradation of PAHs.

A microbial consortium consisted of *Ochrobactrum* sp., *Stenotrophomonas maltophilia* and *Pseudomonas fluorescens*, which was obtained by enrichment culture of sea water samples from China, was shown to degrade various PAHs including BaP [41] (ST20). Further investigation showed that among these strains, *Ochrobactrum* sp. is the only one capable of degrading BaP.

The degradation of BaP has also been demonstrated with the use of a bacterial mixture consisting of *Bacillus cereus* and *Bacillus vireti*. These bacteria derived from an enrichment of the refinery soil contaminated with oil products and enriched in the presence of BaP as a sole substrate. This bacterial mixture was found to degrade almost 50% of 500 mg·L^−1^ of BaP within 35 days [42] (ST21). *cis*-4-(7-Hydroxypyren-8-yl)-2-oxobut-3-enoic acid was identified as a BaP metabolite, however, since two bacterial strains were employed, this metabolite could not be ascribed to a specific strain.

Benzo[a]pyrene degradation has also been investigated using *Rhizobium tropici* CIAT 899 strain, a symbiotic bacterium that was initially isolated from nodules of *Phaseolus vulgaris* plant. This strain could degrade 45% of BaP (from an initial concentration of 60 µg·mL^−1^) within 120 h, in a culture containing yeast extract and mannitol [43] (ST22). Petroleum products have a high imbalance of carbon/nitrogen, therefore, such bacteria are of importance in bioremediation, since they can fix nitrogen, thus providing a better balance of carbon/nitrogen.

Ping et al. reported a BaP degrading bacterial strain, *Klebsiella pneumonia* PL1, following the enrichment of soil samples in the presence of pyrene and BaP. Within 10 days, the strain could degrade almost 56% of 10 mg·L^−1^ BaP, and 63% of 20 mg·L^−1^ pyrene [44] (ST23). Further investigation showed that this strain could also degrade these two HMW-PAHs in paddy soil [44].

Using a petroleum sludge as inoculum, Mishra and Singh isolated 12 bacterial strains following an enrichment in the presence of oil. Among these strains, *Pseudomonas aeruginosa* PSA5 and *Rhodococcus* sp. NJ2 were proven to grow in the presence of BaP as a sole source of carbon. Further investigations showed that these two strains could degrade 88% and 47% of 50 ppm BaP respectively, within 25 days of incubation [45] (ST24). This study also demonstrated that the degradation of BaP was associated with the expression of enzymes involved in aromatic degradations, mainly 2-carboxybenzaldehyde dehydrogenase in *P. aeruginosa* PSA5, and salicylate hydroxylase in *Rhodococcus* sp. NJ2.

Nine bacterial strains were isolated by enrichment in the presence of peptone, beef extract and pyrene, using an inoculum consisting of water and mud samples, collected from various sites (including oil refinery and waste treatment plant). Each of the bacterial strains *Mycobacterium fluoranthenivorans*, *Herbiconiux ginseng*, *Mycobacterium brisbanense*, *Bacillus rnegaterium*, *Sphingobium amiense*, *Mesorhizobium septentrionale*, *Vibrio rumoiensis*, *Olleya* sp. and *Mesoflavibacter zeaxanthinifaciens* could degrade over 60% of 100 µg·mL^−1^ BaP in the presence of salicylic acid within 42 days. Among these strains, *M. zeaxanthinifaciens* showed the best performance in degradation of BaP [46] (ST25). Further investigations showed that this strain could also degrade BaP as the sole source of carbon (25% in 9 days), and that this degradation is improved to approximately 40% (during the same period) in the presence of salicylate.

As discussed in ST16 [37], a consortium consisting of the bacterial strains *Acinetobacter calcoaceticus* MTCC 2409 and MTCC 2289, *Serrati amorcescens* MTCC 2645, *Pseudomonas* sp. MTCC 2445, *Stenotrophomonas xmaltoxphilia* MTCC 2446, and the fungus *Aspergillusterricolavar Americanus* MTCC 2739 was investigated in the context of BaP degradation, when used as a sole substrate. This consortium could degrade 64% of 30 mg·L^−1^ of BaP within seven days, it could also degrade fluoranthene [47] (ST26). Several BaP metabolites were identified, however, because a consortium was used, specific degradation pathways could not be proposed.

All the aforementioned studies have been carried out using soil or sediments as inoculum to isolate bacteria. However, Sowada et al. reported the isolation and characterisation of BaP degrading bacteria from human skin, using swabs from different parts of skin. This enrichment was carried out in microplates in 200 µL, unlike the aforementioned studies, which were generally carried out in bigger volumes, equal or more than 50 mL. Twenty one species of BaP degrading bacteria were identified, and these belong to various genera, including *Micrococcus*, *Bacillus*, *Pseudomonas* and *Staphylococcus*, [48] (ST27). However, in this study, the BaP was dissolved in dimethyl sulfoxide (DMSO) during the enrichment procedures. Therefore, the possibility of DMSO utilisation as a carbon source is not precluded.

### 2.2. Degradation under Extreme Conditions

All the aforementioned studies have been reported in normal conditions which consist of presence of O_2_ (aerobic condition), ambient temperatures of 30–40 °C, low salinity [less than 3% NaCl (wt/v)], neutral pHs (pH ≈ 7). However, pollutant contaminants in general, and PAHs in particular, also occur in extreme environments, characterised by low or elevated temperatures, acidic or alkaline pHs or high salt concentrations. Microorganisms that are capable of growing under such extreme conditions are known as extremophiles; these include halophiles (high salt concentrations), alkaliphiles or acidophiles (high or low pHs, respectively), thermophiles and hyperthermophiles (relatively high and very high temperatures, respectively) and psychrophiles (low temperature). Pollution also occurs in environments with either low levels or absence of O_2_, and microorganisms that thrive is such environments are known as anaerobic. Thus, the exploitation of bioremediation in cleaning such environments requires the use of anaerobic microorganisms or extremophiles. This concept has been a focus of research, and several extremophiles have been reported to degrade pollutants. Readers are referred to the following reviews on petroleum products degradation, including PAHs, by anaerobic microorganisms [19,59,60,61], thermophiles [3,54], and halophiles [62,63,64].

In these studies, bioremediation of aliphatic and aromatic petroleum products has been reported, however, reports on bioremediation of BaP under extreme or anaerobic conditions are scanty; these are summarized in Table 1.

#### 2.2.1. Anaerobic Microorganisms

Under aerobic conditions, the degradation of substrates releases electrons, which are then captured by O_2_ as the terminal electron acceptor; through this process, energy, primarily in the form of ATP, is generated. In anaerobic conditions, molecules or ions other than O_2_ are used as terminal electron acceptors; this includes nitrates or nitrites (producing N_2_, denitrification), sulfates or sulfites (producing S_2_, sulfate or sulfite reducing condition), oxidised ions (leading to reduced ions, ion reducing conditions), and finally, CO_2_ (producing biomethane, methanogenic conditions) [65] Thus, degradation under anaerobic conditions is to be analysed in the context of terminal electron acceptors.

The anaerobic degradation of PAHs containing two to four fused aromatic rings (naphthalene, phenanthrene, anthracene, fluorine, pyrene, and fluoranthene) has been well reported in the last two decades [19,59,60,61]; however, only recently, in 2014, the first report emerged on the anaerobic degradation of BaP using *Pseudomonas* sp. JP1 (ST28) [49]. This strain was isolated following an enrichment experiment of PAHs, including BaP, contaminated sediments and nitrates as the terminal electron acceptor. Under these conditions, this strain could degrade 40% of BaP from media containing 10 mg·L^−1^ BaP within 40 days. The degradations of fluoranthene and phenanthrene were also reported in this study [49]. The addition of maltose and sucrose as growth substrates was associated with an improved rate of BaP biodegradation, while galactose and glucose did not affect this rate.

In another study, a *Microbacterium* sp. M.CSW3 strain was reported to degrade 70% of 0.5 mg·L^−1^ of BaP within 14 days, in the presence of nitrate as a terminal electron acceptor [50] (ST29). BaP degradation rate was augmented in the presence of pyrene and phenanthrene (as co-substrates), while it was inhibited in the presence of glucose and lactose. This study further demonstrated that the use of BaP was associated with the inhibition of enzymes involved in glucose metabolism, which partly explains the reason for the inhibition of BaP degradation by glucose [50]. Using the same strain (*Microbacterium* sp. M.CSW3), Qin et al. tested the degradation of BaP using various BaP/nitrate ratios, and concluded that the BaP/nitrate ratio of 1:33 was associated with the highest BaP degradation rate by *Microbacterium* sp. (ST30) [51]. Another strain, *Cellulosimicrobium cellulans* CWS2 (ST31), was shown to utilise BaP under nitrate-reducing conditions. This strain degraded up to 80% of 10 mg·L^−1^ of BaP in the presence of 1.0 g·L^−1^ of nitrate. The addition of glucose led to an improved degradation efficiency of BaP, unlike the addition of organic acids, such as tartaric acid, which reduced degradation efficiency [52].

#### 2.2.2. Thermophiles

The use of thermophiles in the context of biodegradation has several advantages. For instance, high temperatures improve the bioavailability of molecules (in this instance, BaP), as the result of decreased viscosity, as well as improved mass transfer and kinetics of reactions [3,66]. Thus, these processes, through the use of thermophiles, are expected to mitigate the recalcitrance of BaP to biodegradation.

Zhao and Wong tested the ability of two thermophilic bacteria, *Bacillus subtilis* BUM and *Mycobacterium vanbalaenii*, to degrade BaP at 55 °C. These strains utilise phenanthrene as a sole source of carbon but could not grow in the presence of BaP as a sole substrate. However, in the presence of phenanthrene, *Bacillus subtilis* BUM strain could degrade BaP [53] (ST32). This strain was further tested for its ability to degrade BaP in a soil composting system, and in the presence of phenanthrene, it degraded BaP on BaP-containing soil samples within 42 days [53].

Another evidence of BaP degradation by thermophiles was provided by Feitkenhauer and Markl in 2003. The authors isolated a mixture of thermophilic bacteria *Bacillus* sp. and *Thermus* sp. from a mixture of samples from hot springs, compost piles and industrial wastewaters; they demonstrated that the consortium of these strains could degrade BaP, along with other PAHs, in the presence of hexadecane as a growth factor at 60–70 °C [54] (ST33). Recently, the ability of the strain *Bacillus licheniformis* M2-7 to utilise BaP was investigated at 50 °C. Although no evidence was provided on the ability of this strain to grow in the presence of BaP as a sole substrate, however, phthalic acid was identified as a metabolite, indicating the possibility of biotransformation of BaP by this strain [55] (ST34).

#### 2.2.3. Halophilic Conditions

Arulazhagan et al. reported the ability of a consortium consisting of *Ochrobactrum* sp. VA1, *Enterobacter cloace* VA2, and *Stenotrohpomonas maltophilia* VA3 to degrade benzo[e]pyrene at moderate salinity conditions (3% NaCl, wt/v). This consortium was isolated by enrichment of soil contaminated samples in the presence of phenanthrene [56] (ST35).

To the best of our knowledge, only one study has been dedicated to the evaluation of the biodegradation of a five-ring-containing PAH in relatively high salinity (6% NaCl, wt/v). This study involved the use of the *Ochrobactrum* sp. VA1 (a strain that was initially isolated in the previous study, ST35). This strain failed to grow in a culture medium containing 6% NaCl (wt/v) when benzo[e]pyrene was used as a sole source of carbon [57] (ST36). However, the addition of yeast extract led to the degradation of benzo[e]pyrene, an illustration of co-metabolism.

## 3. General Findings, Gaps and Strategies to Improve the Isolation of Active BaP Degrading Bacteria

### 3.1. Biochemical Metabolic Pathways

The biodegradation pathways of mono-aromatic hydrocarbons (MAHs) and those of PAHs have been extensively studied. In general, this process is initiated with the action of mono- or dioxygenases, which convert these pollutants to dihydroxy aromatic intermediates. Subsequently, a ring cleavage that is affected by the action of intradiol- or extradiol-oxygenases using oxygen takes place. At that point, for MAHs, the generated intermediates join the Krebs cycle. However, in the case of PAHs, this ring opening process will continue in a stepwise fashion, which results in the formation of aliphatic intermediates that will be transformed in Krebs cycle. The biochemical degradation of MAHs and PAHs have been discussed in few reviews [3,20]. Although several studies were devoted on biodegradation of LMW PAHs, probably because of their simple structure and relatively better solubility in aqueous media, fewer studies were reported on BaP. Herein, the degradation pathways of BaP using various bacteria strains are discussed.

#### 3.1.1. Aerobic Catabolism of BaP

As discussed in Section 2 (ST1), the first identified BaP metabolite was reported using *Beijerinckia* B-836 [22]. This strain was shown to transform BaP to *cis*-9,10-BaP-dihydrodiol (Scheme 1), which was confirmed by proton nuclear magnetic resonance (NMR). Another dihydrodiol isomer was identified as 7,8-BaP-dihydrodiol, however it was present at a low concentration, which prevented its stereochemical analysis. *cis*-9,10-BaP-dihydrodiol is the result of the action of a dioxygenase at 9 and 10 positions of BaP, and thus should lead to substituted pyrenes upon ring cleavage (Scheme 1). Schneider et al. used high-resolution mass spectrometry and fluorescence spectroscopy to identify BaP metabolites generated by *Mycobacterium* RJGII-135 (ST6). One of the identified metabolites was 7,8-BaP-dihydrodiol, indicating the action of a dioxygenase at positions 7 and 8 of BaP, which could also lead to the formation of substituted pyrenes. Moreover, a metabolite, which was characterised as either *cis*-4-(8-hydroxypyrene-7-yl)-2-oxobut-3-enoic acid or *cis*-4-(7-hydroxypyrene-8-yl)-2-oxobut-3-enoic acid was identified, indicating a 9,10- or 7,8- ring cleavage respectively (Scheme 1). A metabolite, which is either 7,8-dihydro-pyrene-7-carboxylic acid or 7,8-dihydro-pyrene-8-carboxylic acid, was also identified; further information indicated that BaP ring fission leads to the formation of a substituted pyrene. The metabolite 4,5-chrysene-dicarboxylic acid was also observed using ST6, indicating that the hydroxylation of BaP takes place also at positions 4 and 5 of BaP, leading then to the formation of 4,5-BaP-dihydrodiol. Though 4,5-BaP-dihydrodiol metabolite was not identified, however, the presence of 4,5-chrysene-dicarboxylic acid intermediate supports the ring cleavage at 4,5 positions. These observations show that this strain degrades BaP in more than one route, a phenomena that is frequently reported [20].

Moody et al. used a combination of UV-visible, as well as mass and nuclear magnetic resonance spectroscopic analyses, and identified *cis*-BaP-4,5-dihydrodiol, *cis*-BaP-11,12-dihydrodiol, and *trans*-11,12-BaP-dihydrodiol as vicinal dihydrodiol metabolites of BaP using *Mycobacterium vanbaalenii* PYR-1 [26] [ST4b, Scheme 1]. The identification of *trans*-11,12-BaP-dihydrodiol could be explained by the formation of BaP-11,12-epoxide by cytochrome P450, followed by hydrolysis catalysed by an epoxide hydrolase, which results in formation of vicinal *trans*-dihydrodiol (Scheme 1). In the same study (ST4b), the strain *Mycobacterium vanbaalenii* PYR-1 was shown to generate 4-formylchrysene-5-carboxylic acid and 4,5-chrysene-dicarboxylic acid metabolites when *cis*-4,5-BaP-dihydrodiol was used as substrate. The existence of these metabolites supports a ring cleavage at positions C4 and C5 and at positions C11 and C12. In addition, the metabolite 10-Oxabenzo[*def*]chrysene-9-one was identified in the same study (Scheme 1), which suggests a dihydroxylation at C9 and C10 followed by ring cleavage to produce *cis*-4-(7-hydroxypyrene-8-yl)-2-oxobut-3-enoic acid, though these two metabolites were not identified. This study demonstrated the regio- and stereoselectivity of BaP biotransformation in *Mycobacterium vanbaalenii* PYR-1 strain.

Pyrene-8-hydroxy-7-carboxylic acid and its isomer pyrene-7-hydroxy-8-carboxylic acid were identified as BaP metabolites in *Sphingomonas yanoikuyae* JAR02 (ST15), using tandem mass spectrometric analysis and [^14^C]7-BaP. The existence of these two metabolites supports dihydroxylation of BaP at C9, C10 and C7, C8 (Scheme 1).

The BaP metabolite *cis*-4-(7-hydroxylpyren-8-yl)-2-oxobut-3-enoic acid was also reported when a mixture of *Bacillus cereus* and *Bacillus vireti* were used [42] (ST21). However, since two bacterial strains were employed, one cannot conclude that the metabolite is a result of the action of a single bacterium.

#### 3.1.2. Anaerobic Catabolism of BaP

In relation with anaerobic biodegradation of BaP, so far, metabolites have only been reported in three different studies. In the first one, and using nitrate as an electron acceptor, a bacterial strain of *Pseudomonas* sp. JP1 (ST28) was shown to degrade BaP to generate 1,12-dimethylbenz[a]anthracene and 5-ethylchrysene, which indicate that the ring cleavage of BaP takes place at rings 2 and 3, respectively (Scheme 2). 7,8,9,10-Tetrahydrobenzo[a]pyrene was also identified, which could lead to the formation of pyrene analogs. These metabolites indicate that the degradation of BaP starts by a hydrogenation that takes place at either of the rings 2, 3, or 5, followed by a ring fission.

A landmark study on BaP metabolites identification has been reported by Qin et al. using the strain *Microbacterium* sp. M.CSW3 under anaerobic denitrification conditions in the presence of nitrate as the electron acceptor [51] (ST30). In this study, 4,5-dihydrobenzo[a]pyrene, which results from initial hydrogenation at C4 and C5 of BaP, was identified; this metabolite then undergoes ring cleavage to produce chrysene (Scheme 2). Furthermore, 7,8,9,10-tetrahydrochrysene, which results from hydrogenation of chrysene, was detected. Subsequent ring opening results in the formation of phenanthrene, followed by naphthalene (Scheme 2). In addition, pyrene was detected, an indication that 7,8,9,10-tetrahydrodrobenzo[a]pyrene was formed as initial hydrogenation intermediate. 4,5-Dihydropyrene was also detected, which was further degraded to 4,5-dimethylphenanthrene, 4-methylphenathrene and phenanthrene; the latter can subsequently follow the phenanthrene degradation pathway. At this point, oxidation processes were proposed, as the result of the presence of 3-acetylphenanthrene, and further oxidation to 2-methyl-1-naphthaleneacetic acid, and the latter was converted to naphthalene. Interestingly, the metabolite 1-naphthol, which resulted from the hydroxylation of naphthalene, was subjected to hydrogenation, followed by oxidation and ring opening, as indicated by the presence of 5,8-dihydro-1-naphthalenol,3,8-dihydroxy-3,4-dihydronaphthalen-1(2H)-one and 2-(1-hydroxyethyl)hydroxyl methylbenzene (Scheme 2).

In a recent study, using the strain *Cellulosimicrobium cellulans* CWS2 (ST31), and in the presence of nitrate as electron acceptor, pyrene and phenanthrene metabolites were also identified (as in ST30). In addition, four derivatives of naphthalene: 1-(2-hydroxypropyl)-naphthalene, 1,7-dimethyl-naphthalene, 1-methyl-naphthalene and 2-hydroxy-3-(3-methyl-2-butenyl)-1,4-naphthalenedion, and two derivatives of the MAH benzoic acid (diethyl phthalate and 2-acetyl-3-methoxybenzoicacid), were reported (Scheme 2).

### 3.2. Co-metabolism and Metabolic Inhibition

Co-metabolism has been widely used to improve the degradation of various recalcitrant substrates by microorganisms [18]. This strategy relies on the use of two different substrates, consisting of a growth-promoting substrate (primary substrate), and the non-growth substrate (secondary substrate). Growth substrates are generally less complex and easily utilised by bacteria, while non-growth substrates are recalcitrant to degradation. Thus, when a secondary substrate is used as a sole source of carbon, bacterial growth is slow, even sometimes inhibited, and the addition of a primary substrate promotes the growth, which in turn improves the degradation of the secondary substrate [18].

Careful observation of the afore-discussed studies clearly shows that the degradation of BaP has primarily been carried out when a growth substrate was added in the medium. About 60% of the reported studies on biodegradation of BaP were associated with the use of growth substrates, and the most commonly used were phenanthrene, pyrene, salicylic acid, crude oil, glucose and yeast extract. Obviously, these growth substrates are all less complex than BaP, thus more easily amenable to degradation than BaP. Although the use of growth substrates results in improved BaP degradation, however, this illustrates the difficulties and challenges to isolate bacteria that can grow in the presence of BaP as a sole substrate.

The co-metabolism can also be associated with the opposite effect, that is, the decrease in BaP biodegradation, a phenomenon known as metabolic inhibition. For instance, the addition of succinate in ST15, galactose and glucose in ST28, and glucose as well as lactose in ST29 were associated with a decline in BaP biodegradation. Further investigation in ST29, based on proteomics, showed that the addition of glucose as a growth substrate is associated with inhibition of the expression of enzymes that mediate the degradation of BaP, thus leading to a lower efficiency of BaP degradation [50]. This inhibitory effect of the growth substrate glucose has also been reported on the biodegradation of other PAHs [67]. Thus, it is imperative to investigate the effect of growth-promoting substrates prior to their use in the context of co-metabolism.

### 3.3. Biotransformation, Biomineralisation and Bioaugmentation

Pollutants are used as sources of carbon and energy by microorganisms, and in the aerobic processes, the complete degradation of pollutants leads to the generation of CO_2_, as the ultimate product; this process is known as biomineralisation. However, a microorganism can utilise a pollutant for growth without complete degradation. This process, known as biotransformation, generates intermediate compounds. These intermediates can accumulate in the environment, and some can even constitute secondary pollutants in the environment, causing a new environmental concern [68]. The more complex is the substrate or pollutant, the more difficult is the biominerialisation, and overall, this biomineralisation is favored in the context of the use of microbial consortia. Indeed, an intermediate generated by one bacterial strain can be used as a substrate by another strain, leading to a complete degradation using such consortia [69,70].

In the context of PAHs, BaP is less amenable to biomineralisation than PAHs of lower molecular weights. Nevertheless, five studies (ST4a, ST5, ST8, ST10, and ST15) have reported the generation of CO_2_ from BaP, a clear evidence of biomineralisation. However, a careful observation of these investigations shows that ST4a, ST10, and ST15 involved the use of bacterial consortia, a common feature that has been reported in biomineralisation [70]. Interestingly, in the other two studies (ST5 and ST8), in which single strains were used, biomineralisation was carried out in the context of co-metabolism, with the use of the growth substrate phenanthrene (ST5) and fluoranthene (ST8). Thus, so far, there is no report of BaP biomineraliation that utilises a single bacterial strain and BaP as a sole substrate. Thus, single bacterial strains that can mineralise BaP, in absence of a growth substrate, as the sole substrate awaits isolation and characterisation.

Once isolated and characterised, single bacterial strains can be used in the context of bioaugmentation to improve degradation. This approach rests on the addition of preselected and active microorganisms to a microbial community to enhance the ability of this microbial community to degrade pollutants [3]. This technique has widely been used in the degradation of recalcitrant pollutants [65,71,72,73]. Thus, the isolation and characterisation of more active BaP degrading bacteria will greatly contribute in the application of bioaugmentation strategy.

### 3.4. Use of BaP in Enrichment Experimental Setup

Most of studies discussed in this review involve the use of enrichment approach by incubating inoculums in mineral culture media containing one or more selected compound(s) as the carbon source. Since the aim is to isolate bacteria that can degrade a given molecule, it is conceivable that the same molecule be used as the sole source of carbon in enrichment experiments. However, so far, only seven out of 36 studies utilised BaP as a sole source of carbon in enrichment experiments (Table 1). In the remaining studies, enrichment experiments were primarily carried out with other PAHs, mainly phenanthrene, pyrene, fluoranthene, mixture of PAHs or crude oil. Thus, one can conjecture that the main aim of these studies was to isolate bacteria that degrade compounds used in these enrichment experiments, and BaP biodegradation was tested afterwards, but do not reflect attempts made by investigators to isolate BaP degrading bacteria. Therefore, it is likely to argue that more active and efficient BaP degrading bacteria await isolation.

### 3.5. Genomes, Omics and Functional Metagenomics

Studies on the degradation of pollutants rest primarily on the isolation of bacterial strains, prior to analysis. However, this approach is limited because of the challenges associated with isolation of a single bacterial strain capable of degrading pollutants. The evolving techniques in the field of biotechnology such as cloning, metagenomics, proteomics, and functional genomics have permitted the investigation of bacterial ability to degrade pollutants without the necessity of prior bacterial isolation. They have also allowed the screening and isolation of bacteria that carry unique genes capable of improving biodegradation. As it will be discussed below, these concepts have been exploited in the context of degradation of PAHs with lower molecular weight, however, that on BaP awaits investigation.

Key genes encoding enzymes involved in the degradation of a pollutant can be used as probes to screen, identify and isolate bacteria that can actively degrade a pollutant of interest. For instance, as discussed in Section 3.1, the first step in the biodegradation of PAHs in aerobic conditions is the hydroxylation by a dioxygenase, and the genes that code for these enzymes in naphthalene and phenanthrene degradation are *nahAc* and *phnAc*, respectively [74]. These genes have been used as probes to detect bacteria that degrade the two afore-mentioned PAHs in various environments [75]. Likewise, in the context of pyrene degradation, this enzyme is known as aromatic ring hydroxylating dioxygenase (ARHDO), and has already been used as a probe to track the spread of bacterial degrading pyrene [76,77]. This enzyme is also present in BaP-degrading bacteria as shown by the detection of hydroxylated BaP metabolites (ST1, ST4b, and ST6), however, the gene encoding this enzyme has not yet been used to screen and identify BaP degrading bacteria, as it has been the case with genes associated with naphthalene, phenanthrene and pyrene. Thus, this opens up a potential and an interesting research avenue.

An interesting approach to unravel biochemical pathways of pollutant degradation is the use of transcriptomics (protein profiling or proteomics). This technique has been evaluated in the context of PAH degradation, including pyrene [78,79,80], and recently, reports are emerging on the use of this technique on BaP degradation [50,81]. Thus, transcriptomics is a valuable tool to be further utilised in the context of understanding BaP degradation.

As discussed in Section 3.1.2, the mechanisms of degradation of aromatic compounds has also been studied in anaerobic condition, and in the case of mono-aromatic compounds, the ring opening process is mediated by 6- oxocyclohex-1-ene-1-carbonyl-CoA hydrolase, an enzyme encoded by the *bamA* gene [60,82]. Interestingly, this enzyme has been used as a “probe” to identify and map the presence of these aromatic-degrading bacteria in various environments [82,83,84]. This strategy also awaits to be exploited in the context of BaP degradation in anaerobic conditions.

In the aforementioned studies, the “probes” used were selected based on reported gene sequences in single bacterial strains that were cultured in vitro. However, since the majority of microbes cannot be cultured in vitro, one cannot tap on the vast wealth of possible genes that control biochemical pathways of pollutant degradation. However, high-through-put techniques involving metagenomics have now permitted the exploration and exploitation of genome information of uncultured microbes [75,85]. More specifically, using uncultured microbes directly collected from the environment, functional metagenomics can be carried out by cloning large DNA fragments into specific vectors such as fosmids, BACs (bacterial artificial chromosome), cosmids or plasmids. Thereafter, these clones (also known as metagenomic clones) can be transformed into cells, such as *Escherichia coli*, which can then be used in the degradation of pollutants [86,87]. For instance, Vasconcellos et al. screened a library of metagenomic clones (around 5000) for their ability to degrade the aliphatic hydrocarbon hexadecane, using high-through-put approaches [88]. A total of 72 hexadecane degrading clones were identified [88], and among them, two clones were further shown to also degrade phenanthrene and methyl-phenanthrene [89]. Recently, using the same library, clones capable of degrading naphthalene and the *n*-alkanes (dodecane and tetradecane), while producing exopolymers with emulsifying activities, were also identified [90].

Another approach that permits the study of non-cultivable bacteria is stable isotope probing (SIP). This technique is based on the use of substrates that contains a heavy isotope, such as ^14^C, followed by detection of the incorporated isotopic element in bacterial RNA and DNA. This process allows, selectively, the detection and identification of microorganisms (including uncultured microorganisms) that can utilise a given substrate, leading then to link microbial identification with functional biochemical properties [91]. SIP approach has a wide scope of applications including biodegradation [92]. For instance, it has been recently used in the identification of BaP degrading bacteria from forest soil [93].

The combination of the SIP and metagenomics have been proposed and evaluated. One side, metagenomics permits the analysis of all microbial population present in the environment, and the other, the SIP approach allows the detection of metabolically active microbes only. Thus, the combination of these approaches can overcome the limitations due to microbial complexity in metagenomics and permits detailed analysis of active microbes only [94,95]. Recently, an advanced bioinformatics tool, known as MetaSIPSim, aims at improving the analysis of metagenomics and SIP data, has been described [96].

## 4. Concluding Remarks and Future Perspectives

For the last four decades, degradation of PAHs by microorganisms has been well studied, and the degradation pathways well understood. However, most of this work was devoted to degradation of LMW-PAHs. Limited work has been dedicated to BaP, and the primary reason is its recalcitrance to biodegradation, as the result of its high stability. Nevertheless, few studies have been devoted on BaP biodegradation. However, as this review demonstrates, most of these studies have been carried out in the context of either co-metabolism or using bacterial consortia. For instance, so far, there is no report on a single bacterial strain that can biomineralise BaP when used as a sole source of carbon.

Most microorganisms found in Nature are not “cultivable” in vitro, thus limiting their isolation. However, the development of “-omics” sciences (metagenomics, proteomics, transcriptomics) and SIP techniques now overcomes this limitation. These new approaches have been evaluated in the context of aliphatic and PAHs degradation, however, so far, limited work has been dedicated on BaP, thus opening an avenue for new research on BaP biodegradation.

This review has demonstrated that most of BaP-degrading bacteria are in fact mesophilic bacteria. Limited work has been carried out on halophilic and thermophilic microorganisms, and surprisingly, those degrading BaP under acidophilic and alkalophilic conditions have never been reported so far. Likewise, in anaerobic conditions, only nitrate- reducing microbes have been reported, and investigations are needed to study such microorganisms in sulfa reducing- and methanogenic- conditions. Microorganisms growing in these extreme conditions are characterised by unique biochemical features, thus are important to be investigated.

In relation with the biochemical pathway of BaP, this review has also demonstrated that the first and main important metabolites of anaerobic degradation are dihydroxy derivative of BaP, which thereafter follows the ring opening process, as it has been demonstrated in the degradation of PAH of smaller molecular weights [17,18,19,20]. However, the enzymes involved in BaP degradation have not been reported yet. The ability to relate these enzymes with the various biochemical reactions of BaP degradation, through investigation of techniques such as proteomics, will contribute immensely in our understanding of BaP degradation.

In relation with anaerobic degradation, ring hydroxylation, carboxylation and methylation have been reported to be the first steps in biodegradation of aromatic compounds. However, in the case of the BaP, as shown in this review, this first step is the hydrogenation of one ring, followed by ring opening, a feature that is different from that of the aromatic compounds reported so far. Thus, further studies are required to gain more insight on anaerobic degradation of BaP.

The identification of new enzymes with their roles in chemo-, regio-, and enantioselective chemical synthesis could lead to the development of new tools for organic synthesis using biocatalysis, especially with the recent advancements in the biotechnology including directed evolution and protein engineering [97,98,99].

Overall, this review has highlighted the current gaps that still exist in our understanding of BaP biodegradation, and the research avenues that have been proposed will contribute immensely in filling these gaps.

## Data Availability

Data presented in this review manuscript are available in the quoted references.

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
