# Peer review of "Current Status of and Future Perspectives in Bacterial Degradation of Benzo[a]pyrene"

_ijerph, 2020, doi:10.3390/ijerph18010262_

Round 1
Reviewer 1 Report
The introduction to the review is easy to understand. The authors state that they have compiled a literature review on BaP degradation, with an emphasis on bacterial degradation. The review surveys a number of articles and summarizes them succinctly. Figures 1 and 2 serve as a reference to the various degradation products, while Table 1 seems to be intended as a quick reference guide to a select amount of studies.
I find that there are a number of grammatical issues throughout the document that should be addressed. Use of the word “the” for instance is overused (lines 17, 20, 50, 56, among others). I point out some of these in the comments below, but the first portion of the document in particular should be readdressed regarding certain grammar and rewording issues.
Line 19- Clarify the type of stability (do you mean structural, chemical, biological?) Please clarify.
Line 49-not clear what “latter” is referring to.
Line 68- get rid of space after the / (or you can use and/or).
Line 70- italicize Beijerincka.
Line 77- was the bacterial strain identified at that time? If so, identify. If not, leave as is.
Table 1 exhibits a multitude of formatting inconsistencies and errors. In terms of formatting, statements with the same or a very similar meaning should be defined the same way. For example, inoculum for ST2 is listed as “Soil contaminated samples”, while for ST7 the inoculum is listed as “Contaminated soil sample”. It’s two ways to say the same thing, and since tables should present information in a consistent fashion you should choose one way and stick with it. There are several other instances of this (compare the inoculum for ST 4a, 4b, 12, 13, 30 and 31) ; sometimes you state the inoculum “as in ___” and other times you states “same as in ___”. Again, choose one format and stick with it.
For ST4b you write “same as in ST4” yet there is no ST4 in your table, only ST4a and ST4b.
For ST6 you write “same as above”. This is ambiguous since there are multiple studies listed above ST6; which one are you referring to?
For ST5 you designate the origin sample as “from the USA”. However, you don’t designate the country of origin for any of the other samples except in ST10. You should consistently list the country of origin or don’t list it at all. Furthermore, the USA is quite large and soil profiles are extremely diverse; stating a soil sample comes from the USA doesn’t give the reader much useful information. Contrast this to ST10 (an active cattle pasture Gulf region of Texas), which provides somewhat more specific and useful information. You need to decide the level of specificity you want to convey in this Table for the inoculum and stick with it.
For ST8 is it necessary to say “from a previous study” for the inoculum?
For ST9 inoculum you list “contaminated soil which”. This looks like an incomplete statement. Which what?
For multiple studies you sometimes state the nature of the contamination (and sometimes you don’t). Be consistent in how you present this in your table.
For the conditions of microorganism isolation, you note the media used in ST1 but not in any of the others. If this is stating conditions, you should list the media in each case. Otherwise, you should entitle the column as “Enrichment Conditions” (or something similar) and just note how the media was enriched (as you’ve basically done).
For ST29 you define the concentration and function of sodium nitrate. I suggest that you not do this unless you are going to do the same for every study.
For ST6 get rid of “as above” or refer directly to the study in the conditions column.
For ST9 (in the name of strains column) list the other 3 species. You do so in other cases when there are multiple species involved.
In the “Co-metabolite” column of Table 1 you inconsistently use capital letters for the first letter. Be consistent.
Throughout the main results column you need to keep the use of dashes and spaces consistent. It varies throughout the table.
In the results for ST10…is the CO2 isotopic if it came from radiolabelled BaP?
Perhaps the results for ST14 could use a footnote instead of an explanation in the table
Results for ST26 and ST27- you give very specific volumes and concentrations but don’t do this for any of the other studies.
In lines 95-98 you begin to discuss the metabolites of BaP degradation for ST5 and ST6. You mention compounds that are shown in Figure 1, and you may want to reference Figure 1 earlier than you have. The same can be said for lines 152-157 (and so on).
In lines 105, you say grown on fluoranthene. Do you mean some kind of media or soil supplemented with fluoranthene? If so, please specify.
In line 136 and 303 should it be co-metabolism instead of cometabolism?
On line 138 is the CO2 radio-labelled? If so, please indicate this.
In line 154, state the nature of the sample.
In line 229, get rid of "is".
In line 251, reword to- “the first reports emerged”.
In line 320, also provide the abbreviation NMR for nuclear magnetic resonance.
In Figure 1, please fix the bond angles for OH groups coming off the ring (for products of ST4b, ST6, ST21). You will need to twist the alkyl substituent groups to accommodate this, but the bond angles will be correct.
This will likely be fixed in editing, but the caption for Fig 2 needs to be pushed up onto the same page as the figure.
Author Response
The following are the point by point response to the reviewer's comments:
Line 19- Clarify the type of stability (do you mean structural, chemical, biological?) Please clarify.
Our response: Change has been made as “chemical stability”
Line 49-not clear what “latter” is referring to.
Our response: latter has been changed to pollutants.
Line 68- get rid of space after the / (or you can use and/or).
Our response: “replace by and/or”
Line 70- italicize Beijerincka.
Our response: Done
Line 77- was the bacterial strain identified at that time? If so, identify. If not, leave as is.
Our response: No species was identified
Table 1 exhibits a multitude of formatting inconsistencies and errors. In terms of formatting, statements with the same or a very similar meaning should be defined the same way. For example, inoculum for ST2 is listed as “Soil contaminated samples”, while for ST7 the inoculum is listed as “Contaminated soil sample”. It’s two ways to say the same thing, and since tables should present information in a consistent fashion you should choose one way and stick with it. There are several other instances of this (compare the inoculum for ST 4a, 4b, 12, 13, 30 and 31) ; sometimes you state the inoculum “as in ___” and other times you states “same as in ___”. Again, choose one format and stick with it.
Our response: We have crosschecked the table and have removed the inconsistencies.
For ST4b you write “same as in ST4” yet there is no ST4 in your table, only ST4a and ST4b.
Our response: This has been corrected
For ST6 you write “same as above”. This is ambiguous since there are multiple studies listed above ST6; which one are you referring to?
Our response: It is referring to ST5. This has been corrected
For ST5 you designate the origin sample as “from the USA”. However, you don’t designate the country of origin for any of the other samples except in ST10. You should consistently list the country of origin or don’t list it at all. Furthermore, the USA is quite large and soil profiles are extremely diverse; stating a soil sample comes from the USA doesn’t give the reader much useful information. Contrast this to ST10 (an active cattle pasture Gulf region of Texas), which provides somewhat more specific and useful information. You need to decide the level of specificity you want to convey in this Table for the inoculum and stick with it.
Our response: This has been corrected. There is now consistency throughout the table
For ST8 is it necessary to say “from a previous study” for the inoculum?
Our response: This has corrected.
For ST9 inoculum you list “contaminated soil which”. This looks like an incomplete statement. Which what?
Our response: Which has been removed
For multiple studies you sometimes state the nature of the contamination (and sometimes you don’t). Be consistent in how you present this in your table.
Our response: This has been crosschecked and corrected in the Table
For the conditions of microorganism isolation, you note the media used in ST1 but not in any of the others. If this is stating conditions, you should list the media in each case. Otherwise, you should entitle the column as “Enrichment Conditions” (or something similar) and just note how the media was enriched (as you’ve basically done).
Our response: To be consistent, the details of the medium has been removed in ST1
For ST29 you define the concentration and function of sodium nitrate. I suggest that you not do this unless you are going to do the same for every study.
Our response: We have remove the concentrations, however, for anaerobic culture, the electron acceptor needs to be mentioned. It is the electron acceptor that differentiates the various type of anaerobic cultures or respirations. All the other cultures are aerobic, therefore O2 is the electron acceptor.
For ST6 get rid of “as above” or refer directly to the study in the conditions column.
Our response: Done
For ST9 (in the name of strains column) list the other 3 species. You do so in other cases when there are multiple species involved.
Our response: The statement has been corrected. There are in fact 3 strains, VUN 10 001, VUN 10 002 and VUN 10 003)
In the “Co-metabolite” column of Table 1 you inconsistently use capital letters for the first letter. Be consistent.
Our response: We use capital letter because it is the first letter of the title of the column
Throughout the main results column you need to keep the use of dashes and spaces consistent. It varies throughout the table.
Our response: this has been crosschecked and standardised.
In the results for ST10…is the CO2 isotopic if it came from radiolabelled BaP?
Our response: Yes, in the context of complete degradation, some molecule of CO2 will be radiolabeled. But this information was not provided in the initial publication.
Perhaps the results for ST14 could use a footnote instead of an explanation in the table
Our response: As suggested, this information is made as a footnote of the table.
Results for ST26 and ST27- you give very specific volumes and concentrations but don’t do this for any of the other studies.
Our response: To be consistent, the volume and concentration have been removed
In lines 95-98 you begin to discuss the metabolites of BaP degradation for ST5 and ST6. You mention compounds that are shown in Figure 1, and you may want to reference Figure 1 earlier than you have. The same can be said for lines 152-157 (and so on).
Our response: This has been changed. The first mention of the figure (Figure 1) is made in line 321, and the Figure 1 is shown at the end of the paragraph containing this line 321.
In lines 105, you say grown on fluoranthene. Do you mean some kind of media or soil supplemented with fluoranthene? If so, please specify.
Our response: It should be “ in the presence of fluoranthene”. This has been corrected
In line 136 and 303 should it be co-metabolism instead of cometabolism?
Our response: It should be “co-metabolism” and this has been changed throughout the text
On line 138 is the CO2 radio-labelled? If so, please indicate this.
Our response: CO2 was not radio-labelled
In line 154, state the nature of the sample.
Our response: This should be in "culture mineral medium". This has been changed
In line 229, get rid of "is".
Our response: Done
In line 251, reword to- “the first reports emerged”.
Our response: Done
In line 320, also provide the abbreviation NMR for nuclear magnetic resonance.
Our response: Done
In Figure 1, please fix the bond angles for OH groups coming off the ring (for products of ST4b, ST6, ST21). You will need to twist the alkyl substituent groups to accommodate this, but the bond angles will be correct.
Our response: This has been redone, as suggested
This will likely be fixed in editing, but the caption for Fig 2 needs to be pushed up onto the same page as the figure.
Our response: As mentioned, this will fixed during the editing of the final version.
Reviewer 2 Report
The submitted review entitled " Current status of- and future perspectives in- bacterial degradation of benzo[a]pyrene " has discussed the bacterial degradation of benzo[a]pyrene, not only summarized the existing related research, but also given the future perspectives. The topic of this manuscript is quite interesting. The title: The title is suitable to the included study. Abstract: In my opinion, the abstract should focus on “bacterial degradation of benzo[a]pyrene”, not others. Please simplify the background content. The introduction: It is suggested to give a summary in the introduction in the form of graph or table The whole review: This review is well done and it refers to the latest publications. But there is some minor revision need to finish before publication. When there are many narratives, classification may give people a clearer feeling. For example: for 2.1. There are too many contents, please summarize it by category In my opinion, in table 1, the “Mesophiles” should be changed to “aerobic and normal conditions of temperature and salinity” according to the manuscript. Language: The language of the manuscript is good for publication.Author Response
The submitted review entitled " Current status of- and future perspectives in- bacterial degradation of benzo[a]pyrene " has discussed the bacterial degradation of benzo[a]pyrene, not only summarized the existing related research, but also given the future perspectives. The topic of this manuscript is quite interesting. The title: The title is suitable to the included study.
Abstract: In my opinion, the abstract should focus on “bacterial degradation of benzo[a]pyrene”, not others. Please simplify the background content.
Our response: We agree with the reviewer comment, therefore, we have removed the first sentence of the abstract to make it more focus on benzo[a]pyrene.
The introduction: It is suggested to give a summary in the introduction in the form of graph or table The whole review:
Our response: We feel that introduction is not too long, that it is captures the main point of the manuscript. However, in light of the reviewer comment, we have made a figure (Figure3) that summarises main finding and the gaps. From this figure, the reader will have an overview of the main gaps in our understanding of BaP degradation.
This review is well done and it refers to the latest publications. But there is some minor revision need to finish before publication. When there are many narratives, classification may give people a clearer feeling. For example: for 2.1. There are too many contents, please summarize it by category
Our response: We have re-read this section, and it is difficult to categorise it in different small sections since all these studies were carried out in the same condition, which are aerobic and normal temperature and salinity. However, we have presented the data on the chronological basis, as much as it was possible.
In my opinion, in table 1, the “Mesophiles” should be changed to “aerobic and normal conditions of temperature and salinity” according to the manuscript.
Our response: We have changed it in the Table
Reviewer 3 Report
The manuscript, titled "Current status of- and future perspectives in-
bacterial degradation of benzo[a]pyrene " is quite interesting and fits to the fields of the journals thematic.
This review contains essential analyses of key processes of degradation of the key organic toxicant - benzo(a)pyrene n various environments.
Nevertheless, I have to formulate few suggestions and recommendations for current text of the paper with aim to increase quality of text and scientific novelty:
- Introduction – why the natural sources of the PAHs, e.g. from the plant tissues are not considered? Main emphasis is made to anthropogenic sources.
- “In human, BaP is a well-established carcinogen” – why in human, but not in human body?
- Introduction – why no any data provided on concentrations of PAHs in various environments in zonal scale. E.g., in current literature there are many data about level of PAHs content in extreme environments (Antarctica, Arctic… etc.)
- Omics part is not clear in sense of what taxon’s (phylas, generas) etc. could be responsible for biodegradation, current statements are too general.
- High criticism is connected with this statement: “Most microorganisms found in the nature are not “cultivable” in vitro, thus limiting their isolation. However, the development of “Omics” sciences (metagenomics, proteomics, transcriptomics) and SIP techniques now overcomes this limitation. These news approaches have been evaluated in the context of aliphatic and PAHs degradation, however, so far, limited work has been dedicated on BaP, thus opening an avenue for new research on BaP biodegradation”. Thus, if the key microorganisms are not cultivable, how omics techniques can help us to understand the mechanisms of degradation?
- The final remarks – there is no enough conceptual generalization of review, what taxa’s are responsible for halophilic and thermophilic properties? Are this functional is connected with any active current functions of microorganisms?
Author Response
1.Introduction – why the natural sources of the PAHs, e.g. from the plant tissues are not considered? Main emphasis is made to anthropogenic sources.
Our response: As we have mentioned in the introduction, benzopyrene contamination results from oil industry, and also from the incomplete burning of coal, wood (plant) or meat. To the best of our knowledge, the main concern of BaP in plant is that BaP presents in the environment can easily accumulate in plants. Thus, the presence of BaP in the environment is the main source of BaP in plant, and in the manuscript, we have elaborated on the sources of this BaP in the environment.
2. “In human, BaP is a well-established carcinogen” – why in human, but not in human body?
Our response: We have changed to add "human body"
3. Introduction – why no any data provided on concentrations of PAHs in various environments in zonal scale. E.g., in current literature there are many data about level of PAHs content in extreme environments (Antarctica, Arctic… etc.)
Our response: The manuscript is focused on degradation of BaP by bacteria. Although we agree that knowing PAH concentration in various environment is important, however, expanding on the dynamics of BaP will take us away from our main focus.
4. Omics part is not clear in sense of what taxon’s (phylas, generas) etc. could be responsible for biodegradation, current statements are too general.
Our response: The statement looks general because BaP (or PAH) degradation cannot be restricted to one or few taxons.The diversity of bacteria degrading BaP or PAH is known to be high, indeed it is not restricted to one or two taxa. For instance, even some pathogenic bacterial species have been shown to degrade PAH, including BaP.
5. High criticism is connected with this statement: “Most microorganisms found in the nature are not “cultivable” in vitro, thus limiting their isolation. However, the development of “Omics” sciences (metagenomics, proteomics, transcriptomics) and SIP techniques now overcomes this limitation. These news approaches have been evaluated in the context of aliphatic and PAHs degradation, however, so far, limited work has been dedicated on BaP, thus opening an avenue for new research on BaP biodegradation”. Thus, if the key microorganisms are not cultivable, how omics techniques can help us to understand the mechanisms of degradation?
Our response: The main point in this statement is that, indeed, these techniques do not require the culturing of microorganisms. In theory, one single bacterium (present in a contaminated sample) will be enough to carry out these studies, because these techniques involve DNA or RNA amplification, therefore no in-vitro culturing is required.
6. The final remarks – there is no enough conceptual generalization of review, what taxa’s are responsible for halophilic and thermophilic properties? Are this functional is connected with any active current functions of microorganisms?
Our response: The attempt to identify specific taxa associated with BaP (or complex PHA) degradation in halophilic and thermophilic conditions can only be made if there are many reports on BaP degradation in these conditions. However, as discussed in the paper, limited work has been carried out on this topic, therefore there is no available information to propose taxa that are associated with BaP degradation in these extreme conditions.
Round 2
Reviewer 1 Report
My previous comments and suggestions to the authors have been adequately addressed.